# Digital transformation and liquidity creation in commercial banks: Evidence from the Chinese banking industry

**Wen Wen** [1]*, Ying Liang[2]

1 College of Finance Management, School of Economics and Management, Xianyang Normal University, Xianyang, China, 2 College of Economics, School of Economics and Management, Xianyang Normal University, Xianyang, China

* Wendy_W_@outlook.com

## Abstract

This study investigates the impact of digital transformation on the liquidity creation among banks in China, focusing on 127 commercial banks from 2010 to 2021. The research is driven by the observed decline in the real economy function of China's financial services in recent years. To analyze this trend, the study employs a panel fixed effect model and an intermediary effect test model. The findings indicate that the implementation of digital transformation in the banking sector resulted in a significant enhancement in liquidity generation. The intermediary transmission pathways through which digital transformation influences liquidity creation include optimization of loan loss provision, enhancement of risk tolerance, and mitigating financial disintermediation. Furthermore, the research indicates that the relationship between digital transformation and liquidity generation varies across banks operating in different contexts.

## 1. Introduction

In recent years, innovative financial services such as digital payments, online credit and digital risk management- driven by advancements, which rely on digital technology innovation, continue to emerge, enabling the banking industry to improve operational efficiency through digital transformation [1]. In the digital economy era, the digital transformation of the banking industry has offers distinct advantages over traditional operations and management practices in overcoming the limitations of the old financial industry's business model and growth logic. The adoption of digital methodologies for data collecting and analysis by commercial banks has the potential to effectively mitigate information asymmetries, enhance resource allocation efficiency, and leverage digital technology to engage market participants and foster the growth of the tangible economy. The onset of digital transformation has emerged as a crucial catalyst for the advancement of banks in terms of quality. According to the "China Banker Survey Report (2021)", a significant majority of bankers in China, specifically 56.3%, identified "promoting digital transformation" as a strategic goal among the 88 institutions surveyed. The digitalization of banks in the industry has not only been widely recognized, but regulatory authorities have also actively established a favorable legislative environment. China has introduced additional policy measures to facilitate the process of digital transformation. The

**Data availability statement:** All data files are available from the OSF database (accession number(s) https://osf.io/zsf8d/).

**Funding:** 1. Shaanxi Provincial Department of Education Special Research Program project "Digital Finance and traditional finance Coupling Empower Shaanxi Provincial Rural Revitalization Strategy Development" (project No. : 23JK0247); 2. Xianyang Normal University Special Research Fund Project "Research on Digital Financial Resource Allocation Mechanism in the Upgrading of Industrial Structure in Shaanxi Province" (Project No. : XSYK22010). The funders had no role in study design, data collection and analysis, decision to publish, or preparation of the manuscript.

**Competing interests:** The authors have declared that no competing interests exist.

issuance of the Guiding Opinions on *the Digital Transformation of the Banking Industry and Insurance Industry* by the China Banking and Insurance Regulatory in 2022 delineated a clear framework for the banking industry to achieve substantial outcomes through digital transformation by the year 2025. In the same year, the Chinese government released several economic documents that incorporated the digital transformation of the banking sector as part of the digital transformation and upgrading project for important industries. In light of the positive reception from both the industry and regulatory bodies towards the digitalization of banks, it is essential for the academic community to urgently examine the effects of digital transformation on bank development. Accelerating digital transformation has become an unavoidable strategy for commercial banks to improve quality and efficiency and effectively support the real economy.

As China's economy has entered a new normal, the increasing downward pressure on economic performance and the diminishing capacity of the banking system to support the real economy have become undeniable realities. The diverse financial needs of China's real economy remain inadequately addressed, with financing difficulties and elevated costs impeding high-quality economic development. According to modern financial intermediation theory, the liquidity creation function of commercial banks serves as a primary mechanism for supporting the real economy [2]. This indicates that within China's financial system, which is predominantly characterized by commercial banks, the effectiveness of financial services in serving the real economy is primarily reflected in the level of bank liquidity creation. Considering the challenges posed by the novel coronavirus epidemic, China's real economy is grappling with difficulties in debt repayment, financing expansion, and capital turnover, necessitating that banks and other financial institutions provide sufficient liquidity. In response to these issues, regulatory authorities have implemented several policies in recent years, including allowing banks that meet inclusive finance assessment criteria to benefit from a preferential deposit reserve ratio and reducing excess deposit reserves to signal guidance for credit expansion. These measures have alleviated the short-term funding shortages in the real economy; however, this issue remains challenging to resolve substantively. Consequently, research on enhancing bank liquidity creation is not only a critical factor in addressing the financing dilemmas faced by the real economy in the short term but also an essential pathway for China's medium- and long-term financial and economic development.

It is noteworthy that, in recent years, the ongoing integration of digital technology with traditional financial services has enabled banks to systematically transform their product and service models through the application of digital technologies such as big data [3,4]. As information technology increasingly converges with financial services, banks are also beginning to explore new development models within the realm of the digital economy. For instance, in 2023, the total fintech investment among China's six major banks reached 122.822 billion yuan, reflecting a growth of 5.38%. China Construction Bank leveraged digital technology to establish a comprehensive service platform for rural revitalization, providing nearly 100 billion yuan in agriculture-related credit to rural customers by the end of 2023. Meanwhile, China Merchants Bank has enhanced its digital risk management capabilities and efficiency through the development of an intelligent risk control engine, achieving a business intelligence warning coverage rate of 100% and streamlining digital processes that reduced the average mortgage loan processing time from one month to just 2.7 days. Thus, can the digital transformation of commercial banks mitigate the liquidity shortages faced by Chinese commercial banks? If such an effect exists, will it vary among different institutions? Clearly, these questions warrant further investigation.

In view of this, this study uses the unbalanced panel data of 127 commercial banks in China from 2010 to 2021 to empirically study the impact and mechanism of banks' digital

transformation on bank liquidity creation. It has been observed that the implementation of digital transformation in the banking sector results in an enhanced degree of liquidity generation. Our findings indicate that the digital transformation of banks contributes to improving liquidity creation levels, and that the digital transformation of banks can expand profit margins within the scope of risk control by optimizing the provision for loan losses, improving risk tolerance and mitigating financial disintermediation. Moreover, it is noted that the relationship between digital transformation and the generation of bank liquidity varies depending on the external contexts in which commercial banks operate. These external environments include factors such as the advancement of digital infrastructure and the influence of emerging digital financial formats. Additionally, the internal conditions of banks, including resource allocation and bank type attributes, also influence this relationship.

The paper's prospective contributions primarily reflected in the subsequent aspects: First, existing literature has begun to explore the impact of digital transformation of banks on banking operations and the real economy, but the investigation on this issue focuses on the asset side of banks. The index of bank liquidity creation is a comprehensive measure of the effective flow of liquidity through the banking system by the asset side and the liability side of the bank. Therefore, from a liquidity creation perspective, it is more comprehensive to examine the issue that the digital transformation of banks can enhance the efficiency of banks to serve the real economy. Second, the existing literature on the connection between digital transformation and liquidity creation requires further enhancement, both theoretically and empirically. This paper adopts a more comprehensive approach the potential for banks' digital transformation to reduce or increase liquidity creation through multiple transmission channels. At the theoretical level, the paper provides an in-depth analysis of how the digital transformation of banks affects liquidity creation, and at the same time, it conducting a detailed empirical investigation into the internal dynamics between these two factors. Third, given the significant disparities in the development of various types of banks and regional institutions in China's banking system, based on the development status of Chinese banks, this paper examines the impact of digital transformation on liquidity under differing external environments and internal conditions. It provides reliable empirical evidence to assist banks in formulating digital transformation strategies tailored to local contexts across China.

The remainder of this work is structured as follows: the second section provides a concise overview of the relevant literature and puts forward hypotheses. The Methods section explains the data, specification of variables, and methodologies employed in our estimations. Empirical results **section** involves conducting empirical testing on our model and performing a series of stability tests. The Further discussion section delves deeper into the fluctuation of baseline regression across various scenarios. In last section, study summarizes the research findings and proposes countermeasures and recommendations.

## 2. Literature review and research hypothesis

### 2.1. Digital transformation affects bank liquidity creation

In recent years, the ongoing integration of digital technology represented by big data, cloud computing and artificial intelligence with the financial industry has transformed and fostered the innovation of financial products and service models, resulting in major changes in the business behaviors and structures of banks [5,6]. These changes have profoundly influenced on the level of liquidity creation within banks and their effects on the real economy [7].

The generation of bank liquidity is influenced not only by monetary policy, but also by the extent of credit that banks are willing to extend. The enhancement of liquidity creation can be facilitated through the digital transformation of banks, which leverages the advantages

provided by digital technologies. The liquidity creation level of banks is determined by their deposit absorption capacity (on the liability side) and loan supply scale (on the asset side). The digital transformation of banks will affect liquidity creation in commercial institutions from both asset and liability perspectives.

The digital transformation of banks results the integration of digital technology and financial services, particularly from the perspective of bank assets. According to Xie and Wang [8], banks have the potential to leverage digital technology to enhance connectivity with consumers. This enables them to offer a a comprehensive array of products and services tailored to the diverse needs of long-tail client segments. According to Cenni et al. [9], the implementation of digital transformation in banks can facilitate the assessment of borrowers' creditworthiness through the utilization of tax payment records, fund transactions, and social reputation data available on the Internet. This approach can help mitigate the information asymmetry that exists between banks and loan customers, enhance the clarity and efficiency of information transmission, and reduce the need for manual intervention in data management, thereby reducing credit costs, enhance the availability of credit. Conversely, the digitalization of banks significantly impacts on the dissemination of information within banks by equipping them with advanced technology. According to Berg et al. [10] and Tian et al. [11], the implementation of big data credit models and digital supply chain finance models can enable banks to engage in real-time dynamic monitoring of borrowers' production and operational activities. This, in turn, enhances the efficiency of bank loan fund utilization and management. Furthermore, liquidity creation is positively correlated with economic output and growth [12], and its expansion further stimulates the growth of the real economy. Consequently, this facilitates an increase in credit availability for enterprise production and operational activities.

From a bank's perspective, the digitalization of financial institutions has the potential to foster innovation in both products and channels, while also mitigating the competitive pressures posed by alternative digital financial services. Digital technology enables banks to effectively understand the risk preferences and wealth status of various customer groups, facilitating the implementation of intelligent marketing strategies. This enables banks to effectively identify distinct customer segments and offer differentiated deposit and financial product portfolios tailored to the specific needs of different customer segments. Consequently, banks can shift their focus from merely seeking deposits to actively introducing them [13] thereby enhancing customer loyalty and attracting a broader customer base for deposit purposes. Liquidity creation, as defined, involves the augmentation of deposits (liquid liabilities) to enhance banks' capacity to offer non-current assets, thus fostering improved bank liquidity generation. Hence, this study posits the following research hypothesis:

**Hypothesis 1:** The implementation of digital technology in banks will lead to a higher degree of liquidity generation.

## 2.2. Digital transformation affects the path of liquidity creation

There are three reasons why we posit that the digital transformation of banks enhances liquidity creation.

First, digital transformation has the potential to enhance the efficiency of bank loan loss provisioning. Broadly speaking, the credit quality of a bank plays a crucial role in determining the extent of its reserves for potential loan losses. In cases of poor credit quality or anticipated impairment losses, the bank accountant allocates additional loan loss reserves to mitigate the potential impact of liquidity risk in the future. Currently, with the significant growth of China's banking sector, there has been an increase in the accumulation of concealed

non-performing loans. Banks frequently allocate surplus provisions for loan losses beyond statutory requirements as a risk mitigation strategy. However, it is crucial to recognize that an excessive allocation of provisions for loan losses can adversely affect capital adequacy ratios. To comply with capital regulations, banks may reduce the proportion of high-risk assets, thereby restricting their lending capacity and hindering the enhancement of bank liquidity creation [14]. Furthermore, the allocation of additional credit loss reserves entails banks maintaining a greater amount of funds as a precautionary measure against potential risks. This not only results in increased expenses associated with credit operations and a decrease in bank profits, but also induces banks to overestimate credit risks and curtail credit expansion, consequently diminishing the level of liquidity generation within the banking sector [15]. The implementation of digital technology in the banking sector facilitates the provision of cross-temporal services by financial institutions. In the "pre-loan" phase, financial institutions have the capability to systematically gather, assess, and make informed decisions based on standardized data, such as company financial data, tax information, as well as non-standardized data such as social evaluation. Through the mid-loan link, banks could continuously monitor the movement of cash and provide timely alerts to consumers who may be at danger. It has the capability to objectively evaluate the quality of credit, assist banks in establishing loan loss reserves in a scientific manner, and enhances liquidity.

Furthermore, digital transformation has the potential to enhance banks' risk tolerance levels. In the traditional credit assessment system, banks mostly rely on manual processes to evaluate clients' credit. However, the high cost of verifying information hinders banks from delivering liquidity to the real economy [5]. Weighing the advantages against the disadvantages, banks are struggle to identify loan risks while minimizing expenses, and they are unable to get returns that align with the risks involved. Consequently, there is a need to restrict credit, resulting in a reduction in the generation of liquidity. The extensive amalgamation of financial technology (fintech) and banking holds promise for alleviating this challenge. Banks can efficiently and intelligently gather and analyze enterprise information through the application of machine learning and big data technology. This not only leads to a reduction in the average variable costs associated with credit operations but also facilitates accurate loan delivery and enhances the processing capacity of bank soft information. In contrast, the utilization of artificial intelligence technology within the banking sector has the potential to enhance the precision and effectiveness of financial risk forecasting, strengthen the security and stability of the financial system, and alleviate banks' concerns regarding impending threats [16]. The enhancement of banks' anti-risk capabilities and risk tolerance is supported by the extensive integration of banking operations and financial technology. Under the premise of manageable risks, financial institutions may undertake specific risks to achieve higher profits, thereby facilitating the provision of additional liquidity.

Moreover, the process of digital transformation has the potential to mitigate financial disintermediation. The presence of information asymmetry within the credit market prevents loans at the optimal interest rates. This, in turn, triggers a non-Walrasian equilibrium, wherein banks tend to allocate a greater proportion of loans to large enterprises in the credit market. Consequently, this situation presents challenges in fulfilling the loan requirements of small and medium-sized enterprises (SMEs) [17]. The emergence of Internet finance is set to significantly impact the banking industry. Financial disintermediation occurs when bank deposit money is redirected through on-balance sheet business channels that offer higher returns and lower costs. This process involves attracting depositors' funds from both within the banking sector and outside it. In contrast, inside the realm of off-balance sheet operations, banks' wealth management offerings operate independently of the traditional banking system. Furthermore, as the Internet financial system, which relies on third-party payment platforms,

continues to evolve, banks' wealth management funds will be likely to be redirected. The impact of Internet finance on the liquidity creation of banks is evident through the observed effects of financial disintermediation effect. The implementation of digital transformation in the banking sector will enhance the capabilities of traditional banks by integrating features characteristic of Internet finance. This will enable the provision of more suitable financial products and foster innovation in the "disintermediation" approach to value creation. Consequently, it will attract previously diverted funds back to banks, thereby reinforcing the foundation for bank liquidity creation. Ultimately, this will generate additional liquidity to support the growth of the real economy. This study proposes the following research hypotheses based on the preceding analysis:

**Hypothesis 2:** Digital transformation promotes bank liquidity creation through reasonable provision for loan losses.

**Hypothesis 3:** Digital transformation promotes bank liquidity creation by improving bank risk tolerance.

**Hypothesis 4:** Digital transformation promotes bank liquidity creation by mitigating financial disintermediation.

## 3. Data, variable definitions and methodology

### 3.1. Data

This paper analyzes the annual data of Chinese commercial banks from 2010 to 2021. The primary sample data is refined to exclude policy banks, foreign banks, and bank samples with missing main variables. Continuous variables at the bank level are identified at the 1% quantile. The final sample consists of 127 banks. The primary sources of bank-level data include the BankScope database, supplemented by information received from banks' annual reports. Additionally, macroeconomic data is sourced from the *National Bureau of Statistics website* and the *Wind database*.

### 3.2. variable definitions

**3.2.1. Digital transformation of banks (Tdi).** The measurement methods employed in the existing literature to assess banks' digital transformation encompass a variety of approaches, including the analysis of annual report text [18], the establishment of strategic partnerships with financial technology firms [19], and the quantification of patents related to financial technology [20]. The implementation of these regulations has facilitated further investigation into the digital transformation of financial institutions. Nevertheless, the measures predominantly rely on a solitary data source, thereby presenting challenges related to inadequate characterization or limited sample coverage.

The Digital Finance Research Center at Peking University has developed a comprehensive index system to address the limitations associated with single-source channels in the digital transformation of commercial banks. The index categorizes the process of digitalization in banks into three distinct dimensions: strategic digitalization, business digitalization, and management digitalization. Subsequently, it is calculated the subdivision index for each dimension. The overall index of digital transformation is derived through a weighted synthesis of each dimension index using principal component analysis, as described by Xie and Wang [8] and presented in Table 1. The index considers not only willingness to actively pursue change, but also the tangible outcomes of such transformation in relation to their organizational structure and business kinds. This comprehensive approach allows for a more thorough depiction of the digital transformation of banks. Consequently, this study employs this metric to evaluate the extent of digital transformation within the banking sector.

**Table 1. Digital transformation index system of commercial banks.**

| Primary index | Weight assigned to the primary index | Secondary index | Weight assigned to the Secondary index | Quantification and delineation of index |
|---|---|---|---|---|
| Strategic digitization | 14.89% | frequencies of the utilization of Digital technology | 100% | The frequency of digital technology-related keywords per 10,000 words within the main body of a bank's annual report. |
| Business digitization | 31.22% | Digital channel | 42.22% | Mobile banking, WeChat banking = 1, otherwise 0. |
| | | Digital product | 47.18% | Internet finance, Internet information and e-commerce products = 1, otherwise 0. |
| | | Digital research and development | 10.6% | Total number of digital technology patents in 3 years (logarithm). |
| Management digitization | 53.88% | Digital architecture | 20.84% | The establishment of digital finance related departments and fintech subsidiaries = 1, otherwise 0. |
| | | Digital talent | 28.6% | The proportion of directors with backgrounds in information technology on the board. |
| | | | 28.21% | The proportion of executives with backgrounds in information technology in the executive team. |
| | | Digital collaboration | 22.35% | Investment and cooperation with technology companies = 1, otherwise 0. |

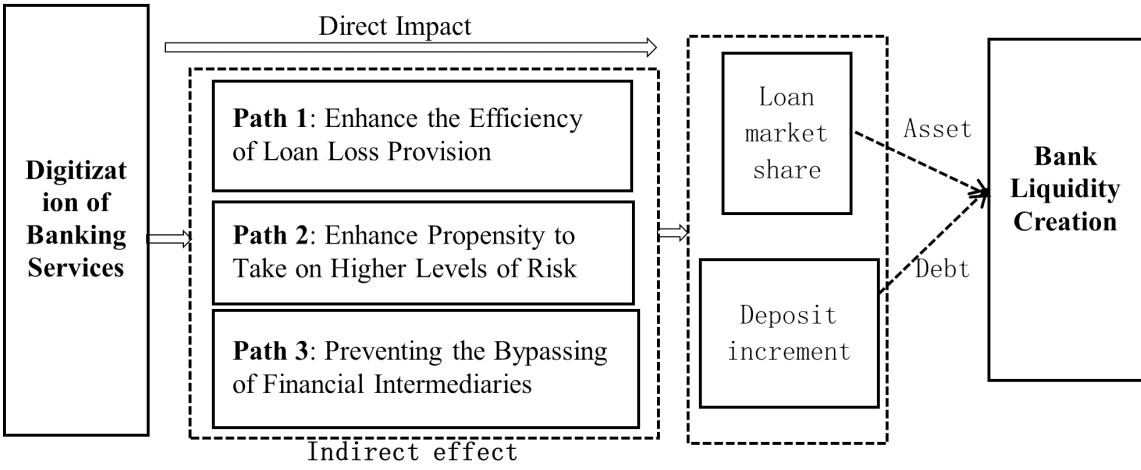

**Fig 1. How the digital transformation of banks affects liquidity creation.**

The trends of each index from 2010 to 2021 are illustrated in Fig 2. Overall, the digital advancement of China's commercial banks has experienced significant growth across all dimensions. The digital transformation of Chinese banks reveals two distinct acceleration points, as shown in Fig 2. After the year 2012, the emergence of Alibaba's Yu 'e Bao exerted a considerable influence on the traditional banking sector, compelling institutions to pursue digital transformation. Furthermore, with the implementation of a series of national digital transformation policies in 2018, the pace of banks' digital transformation is anticipated to accelerate. From a sub-index perspective, it is evident that the strategic digitalization index consistently maintains a high level, underscoring the pivotal role of strategy in the digital transformation of banks.

**3.2.2. Liquidity creation level (Lc).** The three-step cat-fat categorization approach established by Berger and Bouwman [2] is widely employed in the existing literature to

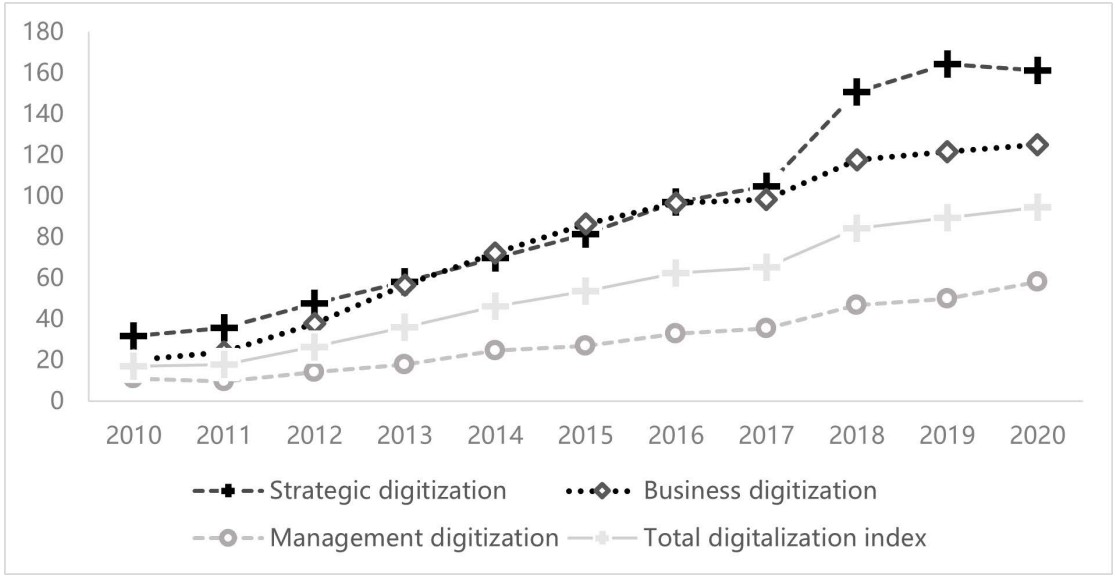

**Fig 2. Time trend of digital transformation of commercial banks from 2010 to 2021.**

develop the bank liquidity generation index system. (1) The balance sheet and off-balance sheet accounts of banks are classified into three groups based on factors such as ease of fund acquisition, transaction cost, and maturity periods. These categories include liquid, semi-liquid, and illiquid accounts, as indicated in Table 2. Appropriate weights should be assigned to each category. According to the concept of liquidity generation, the conversion of a bank's illiquid assets into liquid liabilities results in an increase in the provision of liquidity to external entities. This principle is also extending to off-balance sheet activities. Hence, the generation of liquidity is directly proportional to the magnitude of the bank's illiquid assets, liquid liabilities, and illiquid off-balance sheet activities, with a weighting factor of 1/2. Liquidity is assigned a weight of -1/2 when there are greater amounts of liquid assets, illiquid liabilities and equity, and liquid off-balance sheet operations. The weight of 0 is applied to other semi-liquid accounts. The final bank liquidity creation index is derived by summing the weighted accounts according to their respective weights. The computational approach employed in this study is as follows:

$$
\begin{aligned}
\text{Bank liquidity creation} = &\ 1/2 \times \left( \text{illiquid assets} + \text{liquid liabilities} + \text{illiquid off - balance sheet business} \right) \\
&+ 0 \times \left( \text{semi - liquid assets} + \text{semi - liquid liabilities} + \text{semi - liquid off - balance sheet business} \right) \\
&- 1/2 \times \left( \text{semi - liquid liabilities} + \text{liquid assets} + \text{liquid off - balance sheet business} \right)
\end{aligned}
\tag{1}
$$

To further investigate the variations in liquidity creation from different levels, this calculation method is employed to derive on-balance sheet liquidity creation (Lc-bn) and off-balance sheet liquidity creation (Lc-bw). To mitigate the interference of scale effect, each index is expressed as unit assets multiplied by 100.

**3.2.3. Intermediary variables.** The measurement of bank loan loss provisions (Llp) is based on the methodology established by Tian and Li [15], which the ratio of loan loss reserves to total loans is employed. Risk-weighted assets serve as a metric for evaluating the level of risk undertaken by banks, thereby reflecting their decision-making processes in response to external pressures. This study employs the methodology proposed by Delis and Kouretas [21] as a reference point, and to assess banks' risk tolerance by calculating the

**Table 2. Descriptive statistics.**

| Variable | symbol | Method of measurement | Mean value | Standard deviation | Minimum value | median | Maximum value |
|---|---|---|---|---|---|---|---|
| liquidity creation | Lc | Constructed according to the three-step cat-fat classification method proposed by Berger and Bouwman [2] | 54.345 | 17.421 | 24.078 | 51.436 | 192.160 |
| Digitization of Banking Services | Tdi | Peking University Digital Transformation Index of Chinese commercial banks | 52.137 | 38.468 | 0 | 52 | 200 |
| Provision for loan losses | Llp | Loan loss reserve/total loans | 0.033 | 0.937 | 0 | 0.051 | 0.152 |
| Risk tolerance | Risk | Risk-weighted assets/total assets | 0.587 | 5.342 | 0.223 | 0.649 | 0.915 |
| Financial disintermediation | Findis | (regional corporate bonds + non-financial corporate stocks)/social financing scale | 0.156 | 3,552 | 0.375 | 0.147 | 0.08 |
| Industrial structure | Ind | Gross secondary industry product/Gross regional product | 46.743 | 9.451 | 32.762 | 48.702 | 81.328 |
| Business cycle | M2 | M2 growth rate | 11.460 | 2.346 | 6.990 | 11.010 | 18.950 |
| Level of regional economic development | Gdpg | Regional GDP growth rate | 10.768 | 0.654 | 9.299 | 10.739 | 12.123 |
| Non-performing loan ratio | Npl | Non-performing loans/total loans | 1.627 | 0.764 | 0.049 | 1.214 | 4.376 |
| Capital adequacy ratio | Tcr | Total capital/risk-weighted assets | 13.449 | 3.135 | 0.018 | 12.771 | 57.610 |
| Asset size | Size | Ln (Total assets) | 25.674 | 1.529 | 22.391 | 25.395 | 31.138 |
| Percentage of non-interest income | Nir | Non-interest income/operating income | 34.216 | 10.159 | 0 | 33.315 | 206.077 |
| Profitability | *Roe* | Net profit/net assets | 15.172 | 6.383 | 0.813 | 13.624 | 32.469 |

ratio of risk-weighted assets to total assets. The measurement of financial disintermediation (Findis) is conducted using the financing structure ratio approach proposed by Ji and Li [22]. This method involves dividing the total value of regional corporate bonds and non-financial corporate stocks by the scale of social financing.

**3.2.4. Control variables.** To account for the potential influence of additional variables on the reliability of empirical findings, this paper selects a series of control variables, including bank size, profitability, non-performing loan ratio, income diversity, regional economic development level, broad money growth, industrial structure, etc... Furthermore, considering the potential impact of individual factors, temporal trends, and regional environmental influences, the regression model incorporates controls for the individual effects (Bank), yearly effects (Year), and regional effects (Prov).

### 3.3. Methodology

To evaluate the impact of banks' digital transformation on liquidity generation, this study develops a methodology for conducting a benchmark regression analysis.

$$Lc_{i,t} = \alpha_0 + \alpha_1 Tdi_{i,t} + \sum Controls + \sum Bank + \sum Year + \sum Prov + \varepsilon_{i,t} \tag{2}$$

Within this context, Lc represents the extent to which banks generate liquidity, Tdi indicates the level of digital transformation undertaken by banks, Controls encompasses the collection of control variables, and Bank, Year, and Prov reflect the fixed effects at the bank, annual, and regional levels, respectively. The regression coefficient of $\alpha_1$, which is the primary focus of this study, indicates a statistically significant positive relationship. This finding suggests that the digital transformation of banks has the potential to enhance the generation of liquidity,

thereby aligning with the theoretical expectations posited in this research. This paper develops an intermediary effect testing model based on the benchmark model (2) to examine the presence of three influence channels, namely loan loss reserves, risk tolerance, and financial disintermediation, in the liquidity creation process influenced by banks' digital transformation. The model is constructed using the intermediary effect testing methodology.

$$Mediator_{i,t} = \alpha_0 + \beta_1 Tdi_{i,t} + \sum Controls + \sum Type + \sum Year + \sum Prov + \varepsilon_{i,t} \tag{3}$$

$$Lc_{i,t} = \alpha_0 + \gamma_1 Tdi_{i,t} + \gamma_2 Mediator_{i,t} + \sum Controls + \sum Type + \sum Year + \sum Prov + \varepsilon_{i,t} \tag{4}$$

Mediator refers to intermediary variables, including loan loss reserves (Llp), Risk tolerance (Risk) and financial disintermediation (Findis).

Models (2), (3) and (4) collectively constitute the equations for the intermediation effect test. Generally, three steps are employed to test the intermediate variables. Taking the mediating effect of testing risk tolerance as an example, the first step is to conducting an empirical estimation of model (2) to evaluate the overall impact of banks' digital transformation on liquidity creation without considering risk tolerance. If $\alpha_1$ is significant, it suggests the existence of the overall effect of digital transformation on liquidity creation exists. In the second step, model (3) is estimated empirically. If the coefficient $\beta_1$ is significantly positive, it indicates that the digital transformation will improve the bank's risk tolerance; if the coefficient $\beta_1$ is significantly negative, it indicates that the digital transformation diminishes the bank's risk tolerance, and then the following analysis will be carried out; otherwise, the test is terminated. The third step is to conducting an empirical estimate of model (4) to examine whether the impact of bank digital transformation on liquidity creation changes with the introduction of bank risk tolerance factor. If $\alpha_1$ is not significant, it indicates the absence of an intermediary effect. If $\alpha_1$ is significant, and $\beta_1$ and $\gamma_2$ are also significant, this indicates that there is an intermediate effect. If $\alpha_1$ is significant while $\gamma_1$ is not, it indicates a complete mediation effect. If $\gamma_1$ and $\gamma_2$ coefficients are both significant, it indicates that there is a partial intermediary effect. If $\beta_1$ and $\gamma_2$ are not completely significant, then the Z-statistic constructed by Sobel method can be further tested. If the null hypothesis is rejected: The regression of model (3) is carried out based on model (2). If the coefficient β_1 is significantly positive, it indicates that digital transformation will improve the risk tolerance of banks, and then the following analysis will be carried out; otherwise, the test will be terminated. Subsequently, model (4) is estimated. If the coefficients $\gamma_1$ and $\gamma_2$ are both significantly positive, it indicates that the digital transformation plays a partial mediating role in influencing the risk tolerance of banks in the process of liquidity creation. If $\gamma_1$ is not significant and the $\gamma_2$ coefficient is significantly positive, it indicates that the risk tolerance plays a complete mediator. If the coefficients $\gamma_1$ and $\gamma_2$ are not significant, then there is no intermediary effect. The intermediation effect test of loan loss provision and financial disintermediation follows the same logic as described above.

To account for potential heteroscedasticity and serial correlation, all regression coefficients are clustered and yielding robust standard errors.

### 3.4. Descriptive statistics

Descriptive statistics for the primary variables are summarized in Table 2. The calculated average value of Lc is 0.543, with a range extending from 0.241 to 1.922. This range indicating significant variations in the generation of liquidity across various banks. The mean Tdi value of 52.137 closely corresponds with the data presented in Fig 1. However, the remaining variables fall within an acceptable range.

## 4. Empirical results

### 4.1. Benchmark regression test

Utilizing the provided data and model, we perform an empirical analysis to investigate the impact of banks' digital transformation on the production of liquidity. The results of this analysis are presented in Table 3.

 **4.1.1. Testing the impact of digital transformation on the creation of full-dimensional mobility.** Table 3 presents the initial regression findings concerning the influence of banks' digital transformation on the generation of liquidity. Column (1) considers only the digital transformation of banks as a regression variable, and column (2) adds control variables. The results show that there is a significant positive correlation between digital transformation and liquidity creation, and suggesting that increased bank liquidity is associated with the process of digital transformation of banks. Our results align with existing research indicating that the integration of digital technology and finance can assist banks in expanding their long-tail customer base [23], increase loan volume

**Table 3. Baseline regression test.**

| Variable | Principal regression | | | | Partial regression | |
|---|---|---|---|---|---|---|
| | Lc | | | | Lc_asset | Lc_lia |
| | (1) | (2) | (3) | (4) | (5) | (6) |
| Tdi | 0.043*** | 0.072*** | | | 0.773*** | 0.019** |
| | (2.93) | (3.85) | | | (4.03) | (2.56) |
| L1. Tdi | | | 0.086** | | | |
| | | | (2.35) | | | |
| L2. Tdi | | | | 0.711** | | |
| | | | | (2.16) | | |
| Ind | | -3.041* | -4.012 | -7.027 | -4.038 | -0.025* |
| | | (-1.71) | (-0.51) | (-1.25) | (-1.33) | (-1.65) |
| M2 | | 0.342** | 0.442* | 0.603* | 0.478** | 0.353* |
| | | (2.12) | (1.80) | (1.76) | (2.05) | (1.72) |
| Gdpg | | 0.265*** | 0.242* | 0.284** | 0.338* | 0.329* |
| | | (2.77) | (1.76) | (1.99) | (1.91) | (1.75) |
| Npl | | -1.419 | -1.308* | -1.354 | -0.247* | -0.401 |
| | | (-0.77) | (-1.67) | (-1.12) | (-1.78) | (-0.96) |
| Tcr | | -0.073* | -0.292 | -0.379* | 0.068 | -0.053* |
| | | (-1.95) | (-0.88) | (-1.79) | (1.56) | (-1.78) |
| Size | | 3.086** | 3.073** | 2.065* | 3.069*** | 3.121*** |
| | | (2.25) | (2.11) | (1.78) | (3.12) | (2.79) |
| Nir | | -0.027 | -0.099** | -0.115* | -0.083 | -0.134 |
| | | (-1.25) | (-2.23) | (-1.78) | (-1.52) | (-1.63) |
| Control variable | No | Yes | Yes | Yes | Yes | Yes |
| Individual effect | Yes | Yes | Yes | Yes | Yes | Yes |
| Time effect | Yes | Yes | Yes | Yes | Yes | Yes |
| Regional effect | Yes | Yes | Yes | Yes | Yes | Yes |
| N | 934 | 934 | 934 | 934 | 934 | 934 |
| R² | 0.449 | 0.376 | 0.324 | 0.305 | 0.353 | 0.421 |

Note: ***, ** and * indicate significant at 1%, 5% and 10% levels respectively (two-tailed test); the t-values are presented in parentheses; We employ heteroscedasticity adjusted robust standard error clustering at the bank level. Limited by space, the following tables only report the results of core variables without special instructions, which are the same as here and will not be repeated.

[3,24] and enhancing bank credit supply, thereby increasing bank liquidity creation. Consequently, Hypothesis 1 is supported.

The regression results incorporating lagged explanatory variables are presented in columns (3) and (4). The findings reveal that there is a statistically significant positive relationship between the regression coefficients of one-stage and two-stage lags lag in banks' digital transformation, with a significance level of 5%. Upon further examination, it is evident that the regression coefficient associated with the digital transformation of banks in current is smaller than that observed in the lag period 1. Furthermore, the regression coefficient in lag period 1 is smaller than that observed in the lag period 2. The findings indicate that the influence of banks' digital transformation on liquidity generation does not exhibit a diminishing pattern as the lag period increases. Over time, the digitalization of banks will consistently influence the generation of liquidity, resulting in a delayed effect. Additionally, the improvement derived from digital transformation are persistently reinforced.

**4.1.2. Testing the impact of digital transformation on the creation of sub-dimension mobility.** The impact of banks' digital transformation on liquidity generation is detailed in columns (5) and (6) of Table 3. These columns present insights into two sub-dimensions of liquidity creation: the asset side and the liability side. The findings indicate that digital transformation has the potential to positively impact banks' optimistic market expectations, their willingness to provide credit, and their capacity to generate liquidity on the asset side. Column illustrates how banks utilize digital transformation to enhance service efficiency and broaden their product offerings, thereby attracting more customers, improving storage capacity, and increasing liquidity creation on the liability side. The impact of banks' digital transformation on liquidity generation is evident in both asset and liability dimensions.

## 4.2. Test of the intermediate transmission channel

The aforementioned research and analytical findings indicate that the digitalization of banks exerts a positive influence on the generation of liquidity. This section examines three distinct ways in which the digital transformation can impact the generation of liquidity using the intermediary effect test model. These ways include the provision for loan losses, enhancement of risk tolerance, and mitigation of financial disintermediation.

**4.2.1. Provision for loan losses.** The implementation of digital transformation will bolster banks' confidence in the expected return on investment from loan ventures, expand their profit margins, and reduce the incentive for earnings manipulation. Consequently, banks will be motivated to allocate fewer loan loss reserves and engage in more lending activities within the confines of regulatory obligations, thereby enhancing liquidity creation. The findings of the investigation and analysis are presented in Table 4.

The findings from the mediation effect test of loan loss provision as an intermediary variable are presented in columns (1) and (2) of Table 4. Analysis of model (3), reveals that the Tdi coefficient of the variable in column (1) indicates a statistically significant negative association at the 5% significance level. The adoption of digital technology, commercial banks have enhanced their capacity for monitoring loan credit risks, improve forward-looking provisions for credit losses, and mitigate excessive provisioning for loan losses to cushion risks. This finding indicates a negative correlation between digital transformation and the provision of loan losses. Based on the findings of model (4), it is evident that the Llp coefficient associated with the intermediate variable in column (2) indicates a statistically significant negative relationship at the 5% significance level. This suggests that the decrease in loan loss provisions has a positive impact on the amount of liquidity generation within banks. The intermediary effect test confirms that the digital transformation of banks can reduce provisions for loan losses,

**Table 4. Test of mediating effect.**

| Variable | Llp | Lc | Risk | Lc | Findis | Lc |
|---|---|---|---|---|---|---|
| | (1) | (2) | (3) | (4) | (5) | (6_ |
| Tdi | -0.024** (-2.11) | 0.011*** (2.77) | 0.002* (1.91) | 0.017* (1.75) | -0.048* (-1.91) | 0.064* (1.75) |
| Llp | | -1.264** (-2.05) | | | | |
| Risk | | | | 0.063** (1.98) | | |
| Findis | | | | | | -0.213* (-1.88) |
| Ind | -5.065 (-1.37) | -5.022* (-1.88) | -7.036* (-1.75) | -5.031** (-2.06) | 5.509* (1.74) | 5.016** (2.34) |
| M2 | 0.315** (2.21) | 36.112** (2.45) | 49.209** (2.34) | 34.067*** (2.82) | -38.052* (-1.71) | 38.103* (1.90) |
| Gdpg | 0.232* (1.83) | 0.078* (1.67) | 1.166** (2.11) | 0.105* (1.79) | 0.022* (1.65) | 0.092* (1.73) |
| Npl | 0.319* (1.81) | -0.183 (-0.71) | -0.315* (-1.87) | -0.244 (-0.80) | -0.079 (-1.06) | -0.209 (-1.54) |
| Tcr | -0.354 (-1.14) | -0.113* (-1.66) | 0.218* (1.86) | -0.106* (-1.75) | 0.052* (1.70) | -0.135* (-1.88) |
| Size | 3.160*** (3.12) | 3.055** (2.55) | 2.119*** (3.15) | 3.051** (2.47) | -2.017** (-2.23) | 3.062*** (2.68) |
| Nir | 0.218 (1.44) | 0.119* (1.65) | 0.096* (1.78) | -0.105 (-1.27) | -0.049* (-1.85) | 0.167 (1.62) |
| Individual effect | Yes | Yes | Yes | Yes | Yes | Yes |
| Time effect | Yes | Yes | Yes | Yes | Yes | Yes |
| Regional effect | Yes | Yes | Yes | Yes | Yes | Yes |
| N | 934 | 934 | 934 | 934 | 934 | 934 |
| R² | 0.540 | 0.328 | 0.407 | 0.413 | 0.496 | 0.443 |

ultimately enhancing liquidity creation levels. This indicates that provisions for loan losses serve as an intermediary in the process of improving liquidity generation through banks' digital transformation. Thus, Hypothesis 2 is supported.

**4.2.2. Risk absorption mechanism.** The adoption of big data, artificial intelligence, and other technologies has significantly diminished the knowledge asymmetry between credit parties and enhanced banks' ability to manage future risks through digital transformation. Based on the assumption of manageable risks, the market will incentivize banks to undertake specific risks in pursuit of higher risk-adjusted returns, thereby increasing liquidity. The findings form mediating effect test of risk tolerance as a mediating variable are presented in columns (3) and (4) of Table 4. The results from model (3) indicate that the Tdi coefficient for the variable in column (3) is statistically significant at the 10% level. This indicate that digital transformation has the potential to enhance banks' risk tolerance. This is attributed to the integration of advanced technology in banking operations, which effectively enhances the information transparency of lending enterprises and reduces banks' concerns about unexpected risks. Consequently, under the premise of controllable risks, banks are inclined to pursue greater interest spread returns by taking certain risks voluntarily, which is ultimately reflected in the increase of risk tolerance. Based on the findings of model (4), it is evident that, following the implementation of digital transformation measures, the Risk coefficient associated with the intermediate variable in column (4) exhibits a statistically significant

positive relationship at the 5% significance level. This suggests that an enhancement in risk tolerance will incentivize banks to augment their liquidity. The intermediary effect test confirms that the digital transformation of banks enhances liquidity creation by improving their risk tolerance, indicating that risk tolerance functions as an intermediary in the process through which digital transformation elevates liquidity levels. This finding substantiates the validation of Hypothesis 3.

**4.2.3. Financial disintermediation mechanism.** The digitalization of banks has the potential to redirect funds from emerging financial platforms like Internet finance back towards their core on-balance sheet operations. This, in turn, can significantly increase the volume of loanable funds and deposits held by banks, while facilitating the allocation of more illiquid assets and current liabilities. Consequently, it enhances the overall level of liquidity creation. The findings from the test of the intermediate effect of financial disintermediation as an intermediary variable are presented in Table 4. The analysis of model (3) indicates that the Tdi coefficient for the variable in column (5) indicates a statistically significant negative relationship at the 10% significance level. This suggests that the process of digital transformation may impede financial disintermediation. Because banks leverage digital technology to lower the barriers to financial services, expand loan offerings for small and micro enterprises, agriculture and vulnerable groups, alleviate the financing constraints of these groups, and promote the development of the real economy. Based on model (4), it is evident that in column (6), after the digital transformation is considered, the Findis coefficient of the intermediary variable indicates a statistically significant negative relationship at a significance level of 10%. This finding suggests that preventing financial disintermediation positively impacts bank liquidity generation. In summary, digital transformation can mitigate financial disintermediation, thereby enhancing liquidity creation. This suggests that financial disintermediation serves a partial intermediary role in the process of improving the liquidity creation capacity associated with banks' digital transformation. Hypothesis 4 has been validated.

## 4.3. Robustness test

**4.3.1. Regression model replacement.** The application of regional and temporal bidirectional fixed models is a widely adopted method in regression analysis. However, it is important to note that t while this model provides enhanced flexibility, it may have certain limitations regarding endogenous control. Consequently, the researchers employed the high-order joint fixed effect method as proposed by Moser and Voena [25] as a point of reference. Additionally, the time * region fixed effect was adjusted to reevaluate the model. After accounting for the effects of higher-order joint fixation, the results in column (1) of Table 5 indicate that the conclusions remain robust.

**4.3.2. Replace explanatory variables.** To enhance the reliability of the research findings presented in this paper, the fundamental explanatory factors are substituted. In this study, the digital transformation of banks is represented by the bank-level Fintech index developed by Li and Yang [26]. The index selects fintech-related terms associated with banks from news media. The extent of fintech advancement in banks is quantified by the frequency of specific term occurrences. Currently, despite the rapid growth of the digital economy, there is a notable lack of media coverage regarding the implementation and development of financial technology in banks, even though it has garnered significant attention from various outlets. This study employs the financial technology index as a metric to evaluate the extent of digital transformation within banks, specifically for conducting a benchmark regression analysis. The findings presented in column (2) of Table 5 further substantiate the primary results.

**Table 5. Robustness test.**

| Variable | Replacement regression model | Alternate explanatory variable | Change sample interval | Exclude outliers |
|---|---|---|---|---|
| | (1) | (2) | (3) | (4) |
| *Tdi* | 0.063* | 0.127*** | 0.032** | 0.029* |
| | (1.93) | (2.85) | (2.11) | (1.85) |
| Control variable | Yes | Yes | Yes | Yes |
| Individual effect | Yes | Yes | Yes | Yes |
| Time effect | No | Yes | Yes | Yes |
| Regional effect | No | Yes | Yes | Yes |
| Region * time- joint fixed effect | Yes | | | |
| *N* | 934 | 934 | 827 | 841 |
| *R²* | 0.472 | 0.517 | 0.313 | 0.358 |

**4.2.3. Missing variables.** The COVID-19 pandemic that emerged in late 2019 has significantly impacted micro, small, and medium-sized enterprises (MSMEs) in China, resulting in notable repercussions on bank assets and credit demand. This article intentionally excluded the 2020 samples to mitigate the influence of the pandemic on banks' liquidity creation. Instead, it reevaluated the effect of banks' digital transformation on liquidity creation. Following the exclusion the samples from 2020, the findings in column (3) of Table 5 indicate that the Tdi coefficient remains significantly positive. This suggests that the digital transformation of banks can effectively enhance liquidity generation.

**4.3.4. Delete outlier banks.** The potential existence of systemic disparities between banks that pursue digital transformation and those that do not can be attributed to differences in initial endowments, local government support, and regional infrastructure. To achieve this objective, the banks that have not implemented digital transformation over the specified time frame (banks with a bank digital transformation index of 0) are excluded from the sample. The findings presented in column [4] of Table 5 indicate that the results remain robust even after excluding cases with extreme values.

## 4.4. Endogeneity test

To mitigate potential endogeneity concerns in the model, this study employed the instrumental variable approach and the tendency matching method (PSM) to analyze the empirical analysis findings presented above.

**4.4.1. Instrumental variable method.** This study investigates potential endogeneity concerns in the benchmark model by employing instrumental variables derived from the penetration rate of non-bank fintech [27]. Specifically, this paper focuses on the instrumental variable associated with the penetration rate of non-bank fintech, which is measured using the depth sub-index of Peking University's Digital Financial Inclusion Index. This index assesses users' utilization of digital financial services provided by non-bank technology institutions across six dimensions: payment, credit, and investment. The digital transformation of a bank is more likely to be facilitated by a greater presence of non-bank fintech in the region where the bank is operates. Furthermore, individuals' utilization is not inherently linked to the generation of bank liquidity, thereby meeting the criteria of externality. Therefore, it is appropriate to employ the depth sub-index as an instrumental variable within the context of Peking University's digital inclusive finance.

The results of two-stage least squares (2SLS) estimates, adjusted using instrumental variables, are presented in column (1), (2) of Table 6. In column (1), the Tdi_IV coefficient of the instrumental variable in the initial stage demonstrates a statistically significant positive relationship at the 1% level. This implies that as the penetration rate of fintech increases, there is a substantial enhancement in the degree of digital transformation of banks. Additionally, there is a correlation between the instrumental variables. The results of the second stage regression presented in column (2) indicate that the Tdi coefficient of the variable demonstrates a statistically significant positive relationship at the 1% significance level. This finding suggests that the primary findings remain robust even after addressing endogenous issues such as missing variables and reverse causality by employing instrumental variables.

**4.4.2. Exogenous impact.** The establishment of Internet infrastructure plays a crucial role in shaping the digital transformation of financial institutions. The "Broadband China" strategy and Implementation Plan were announced by The State Council in 2013 with the aim of advancing China's broadband infrastructure. From 2014 to 2016, a total of 121 "Broadband China" demonstration cities were established across the country. Consequently, this study employs a multi-phase differential model (DID) to investigate the impact of digital transformation on liquidity generation, utilizing the "Broadband China" demonstration city pilot as an exogenous factor. Where DID is defined as Treat * Post is defined as a multi-phase differential variable, while "Treat" denotes whether it represents the experimental group. If the city in which the sample bank is located is designated as a pilot city, Treat is assigned a value of 1; otherwise, it receives a value of 0. 'Post' denotes the specific time at which the city hosting the sample bank gains approval to participate in the pilot program. The variable 'Post(n)' represents the time

**Table 6. Endogeneity test.**

| Variable | Instrumental variable test | | Strategic impact of *Broadband China* | |
|---|---|---|---|---|
| | *Tdi* | *Lc* | *Lc* | *Lc* |
| | **(1)** | **(2)** | **(3)** | **(4)** |
| *Tdi* | | 0.107*** (14.79) | | |
| *Tdi_IV* | 42.372*** (25.21) | | | |
| *Treat*Post* | | | 0.895*** (3.87) | |
| *Treat*Post*(-3) | | | | -0.343 (-0.67) |
| *Treat*Post*(-2) | | | | -0.107 (-0.08) |
| *Treat*Post*(-1) | | | | 0.453 (1.10) |
| *Treat*Post*(0) | | | | 0.429* (1.67) |
| *Treat*Post*(+1) | | | | 0.713** (2.33) |
| *Treat*Post*(+2) | | | | 0.726** (2.08) |
| *Treat*Post*(+3) | | | | 0.781** (2.32) |
| Control variable | Yes | Yes | Yes | Yes |
| Individual effect | Yes | Yes | Yes | Yes |
| Time effect | Yes | Yes | Yes | Yes |
| Province effect | Yes | Yes | Yes | Yes |
| N | 934 | 934 | 934 | 934 |
| Kleibergen-Paap rk LM statistics | 55.172 [0.000] | | | |
| Kleibergen-Paap rk Wald F statistics | 68.605 {4.730} | | | |
| $R^2$ | | 0.291 | 0.799 | 0.775 |

period comprising n observations before and after the inclusion of samples from the experimental group or the corresponding control group in the 'Broadband China' demonstration city. This inclusion first occurred in the city where the bank of the treatment group is located, covering a duration of seven years. The specific formulation of the model is as follows:

$$Lc_{i,t} = \alpha_0 + \alpha_1 Treat_t * Post_{i,t} + \sum Controls + \sum Type + \sum Year + \sum Prov + \varepsilon_{i,t} \quad (5)$$

$$Lc_{i,t} = \alpha_0 + \alpha_1 Treat_t * \sum Post(n)_{i,t} + \sum Controls + \sum Type + \sum Year + \sum Prov + \varepsilon_{i,t} \quad (6)$$

The regression results derived from the multi-period differential model is presented in Table 6. Column (3) indicates a statistically significant positive DID coefficient at the 10% level. This finding suggests that the implementation of the "broadband China" strategy has resulted in a notable enhancement in bank liquidity creation. This enhancement may be attributed to the influence of technology spillover effects on the transformation of banks' business models. The fourth column of the analysis investigates the trend characteristics related to the impact of the "broadband China" strategy on the generation of bank liquidity, as evidenced by the time series data. Based on the findings, it can be concluded that the regression coefficients from Treat*Post (-3) to Treat*Post (-1) do not exhibit statistical significance. This indicates that the observed changes in bank liquidity creation between the experimental and control group are consistent with the parallel trend hypothesis. Conversely, the coefficient from Treat*Post (0) to Treat*Post (3) indicates a significant increase of at least 10%, indicating a substantial rise in bank liquidity creation immediately following the implementation of the "Broadband China" strategy. The findings indicate that the conclusion remains strong even when considering the external influence of the "broadband China" policy.

## 5. Further discussion

Based on the data, it is evident that the impact of digital transformation on bank liquidity generation varies across diverse banking contexts. To further explore this diversity, we begin by examining the external and internal conditions faced by banks, and subsequently investigate the influence of digital transformation on the generation of bank liquidity.

### 5.1. External environmental impact of digital transformation

**5.1.1. Infrastructure construction.** Within the context of the digital economy, a widely accepted perspective exists regarding the significance of digital infrastructure. The digitalization of a bank is inherently intertwined with the support provided by the digital infrastructure surrounding. This paper utilizes the ratio of Internet broadband access ports to the number of residents in the bank's location region as a indicator for evaluating the development of digital infrastructure in that region [26]. The samples are classified into two groups based on the median, representing high and low levels of digital infrastructure development. Table 7 presents the group test results in columns (1) and (2). Banks in the group exhibiting a higher level of digital infrastructure development have a higher digital transformation coefficient (0.076) compared to banks in the group with lower digital infrastructure development level (0.024). This indicates that banks with better infrastructure can leverage it to accelerate digital transformation and enhance their liquidity creation. A possible explanation is that in regions with perfect digital infrastructure, banks can capitalize on favorable conditions to promote their own digital transformation, and advance the use of advanced technology to innovate financial products to enhance their competitiveness, thus promoting the creation of bank liquidity.

Table 7.  Influence analysis of heterogeneity: External factors.

| Variable | Level of digital infrastructure development | | Impact of emerging digital financial formats | |
|---|---|---|---|---|
| | high | low | strong | weak |
| | (1) | (2) | (3) | (4) |
| Tdi | 0.076** | 0.024*** | 0.015* | 0.063** |
| | (2.25) | (3.01) | (1.78) | (2.12) |
| Control variable | Yes | Yes | Yes | Yes |
| Individual effect | Yes | Yes | Yes | Yes |
| Time effect | Yes | Yes | Yes | Yes |
| Regional effect | Yes | Yes | Yes | Yes |
| N | 467 | 467 | 467 | 467 |
| $R^2$ | 0.304 | 0.426 | 0.257 | 0.293 |

**5.1.2.  Emerging manifestations of digital finance.**  The emergence of innovative digital financial formats is expected to exert competitive pressure on banks, particularly in relation to the adoption of these formats by consumers. This adoption is expected to have a substantial impact on the traditional loan services offered by banks. The level of development of digital finance varies across different locations, leading to disparate effects on banking institutions. The digital finance index is defined by the ratio of the bank's local branch structure to its digital index, as indicated by the research conducted by Xie and Wang [8]. The Digital Finance Index utilizes the coverage data from Peking University's Digital Inclusive Finance Development Index to evaluate the intensity of a bank's total exposure to emerging forms of digital finance. The sampling process entails dividing the samples into two groups based on the median. Table 7 presents the group test results in columns (3) and (4). In the group exhibiting a strong impact on digital financial formats, the coefficient of digital transformation of banks (0.015) is smaller than that in the group with a weak impact on digital financial formats (0.063). This indicates that the development of digital finance will diminish the positive relationship between the digital transformation of banks and the creation of liquidity. This phenomenon could be attributed to the fact that the development of digital finance, while improving the efficiency of financial services and stimulating the digital transformation of commercial banks, will fosters the emergence of additional financial service platforms competing for market share in credit provision, creating a new competitive landscape, and reduce the credit market share of commercial banks.

## 5.2.  The impact of internal factors on the process of digital transformation

**5.2.1.  Endowment of resources in a bank.**  The relationship between banks' digital transformation and liquidity creation may be influenced by their own resource endowments. The banking industry, characterized by its abundant resources, exhibits a high level of competitiveness in terms of organizational structure, brand recognition, and talent acquisition. Consequently, there is a increasing tendency to explore innovative business models as a means to identify new for profit growth. Furthermore, according to Xie et al. [28], banks that possess substantial resource endowments are more inclined to pursue digital transformation initiatives through independent research. In contrast, banks with limited resource endowments tend to depend on the support of fintech companies to effectively achieve their digital transformation goals. Banks that possess abundant resources experience more pronounced technological spillover effects resulting from digital transformation compared to banks with limited resources. According to Sheng et al. [29], financial

institutions possessing robust resource endowments are more proficient at leveraging the opportunities presented by digital technologies and taking a leadership role in the process of digital transformation. This study evaluates banks' resource endowment through asset scale measurement, and categorizes the samples into two groups based on the median, based on the median: high and low resource endowment. The findings presented in columns (1) and (2) of Table 8 indicate that banks possessing greater resource endowments demonstrate a superior advantage in the context of digital transformation, thereby facilitating enhanced levels. This may be attributed to the fact that banks with high resource endowments typically have higher levels of capitalization, are better positioned to leverage fintech for business transformation and product innovation, and have strong borrowing capacity, thus generating increased liquidity [14].

**5.2.2. Bank classification.** Urban commercial banks and rural commercial banks encounter disadvantage in terms of resource endowment, technological capabilities, human capital, and social awareness when compared to large state-owned commercial banks and joint-stock commercial banks. These disadvantages impede the advancement of digital financial technology. Furthermore, the financial technology innovations introduced by prominent state-owned banks and joint-stock commercial banks has imposed constraints for urban commercial banks and rural commercial banks in terms of market expansion. Additionally, these institutions are witnessing a decline in their initial market shares, thereby diminishing the positive effects of digital transformation compared to their counterparts within the same categories."

This article categorizes the sample banks into four groups, namely large state-owned banks, joint-stock banks, urban banks, and rural banks, in order to evaluate the differential effects of digital transformation on liquidity creation across different types of banks. Table 8 presents the group test results in columns (3) to (6). The digital transformation of big state-owned banks and joint-stock banks exerts a more pronounced impact on encouraging liquidity generation as compared to urban commercial banks and rural commercial banks. This may be attributed to the credit structure of large state-owned banks and joint-stock banks which is characterized by concentrated customers, high proportion of large loans, and systematic business development. With the support of digital technology, large banks have significant competitive advantages in marketing, risk assessment and other areas. By reducing operating costs associated with bank credit operations through economies of scale the willingness of banks to invest liquidity in the medium and long term, thereby improving levels of liquidity creation [30].

**Table 8. Analysis of heterogeneity: internal factors.**

| Variable | Resource endowment | | Type of bank | | | |
|---|---|---|---|---|---|---|
| | more | less | Large state-owned bank | Joint-stock commercial bank | City commercial bank | Rural commercial bank |
| | (1) | (2) | (3) | (4) | (3) | (4) |
| Tdi | 0.081*** | 0.023** | 0.079* (1.92) | 0.084* (1.85) | 0.016* (1.67) | 0.013* (1.72) |
| | (2.77) | (2.35) | | | | |
| Control variable | Yes | Yes | Yes | Yes | Yes | Yes |
| Individual effect | Yes | Yes | Yes | Yes | Yes | Yes |
| Time effect | Yes | Yes | Yes | Yes | Yes | Yes |
| Regional effect | Yes | Yes | Yes | Yes | Yes | Yes |
| N | 467 | 467 | 51 | 120 | 512 | 251 |
| R² | 0.431 | 0.448 | 0.327 | 0.313 | 0.296 | 0.211 |

## 6. Conclusions

In the context of slowing economic growth exacerbated by the impact of the novel coronavirus pandemic, addressing the liquidity shortage faced by China's commercial banks has become a critical component in enhancing the transmission mechanism of monetary policy and ensuring that the financial system effectively supports the real economy. Building on this foundation, this paper first examines the specific mechanisms through which banks' digital transformation influences their liquidity creation levels. It empirically analyzes data from 127 commercial banks in China spanning the years 2010 to 2021, ultimately drawing the following key conclusions: First, an increase in the level of digital transformation among banks generally enhances their liquidity creation capacity, thereby facilitating a more effective support of the real economy by the financial system. This conclusion remains robust even after controlling for endogeneity and conducting various robustness tests. Second, the results of the transmission channel tests indicate that banks' digital transformation can optimize loan loss provisions, enhance their risk tolerance, and mitigate financial disintermediation. Consequently, banks are able to expand profit margins within the confines of risk management, thereby improving their liquidity creation levels. Third, from an external environmental perspective, the positive impact of banks' digital transformation on liquidity creation is more pronounced in regions with superior infrastructure development and a lesser influence from emerging digital financial formats. In regions with robust digital infrastructure, commercial banks can leverage favorable conditions to facilitate their digital transformation and enhance competitiveness, thereby promoting liquidity creation. In areas where the influence of emerging digital financial formats is limited, the digital transformation of commercial banks serves as a more effective alternative to traditional credit models, further fostering liquidity generation. From the perspective of internal conditions, the positive impact of digital transformation on liquidity creation is more pronounced in banks with substantial resource endowments, particularly large state-owned and joint-stock banks. This is attributable to their advanced technological capabilities in both hardware and software, as well as their ability to leverage superior channel resources and information security advantages. Consequently, these institutions are better positioned to capitalize on the scale benefits afforded by digital transformation, thereby enhancing their levels of liquidity creation.

Building on the conclusions, we propose several policy recommendations: First, the liquidity shortage faced by banks has impeded the effective transmission of monetary policy to some extent. The digital transformation of banks can enhance their liquidity levels; therefore, regulatory authorities should encourage commercial banks to advance their digital transformation processes. This can be achieved through policies that promote a deep integration of fintech with commercial banking and the real economy. Such policies should include tax incentives and government subsidies aimed at directing national and market resources toward the digital innovation of banks. Additionally, regulatory authorities should foster the enthusiasm of banks and other financial institutions for research and development (R&D) innovation. Furthermore, they should facilitate constructive interactions among industry, academia, and research entities to promote the accumulation and transformation of digital technological advancements within the commercial banking sector. The primary objective is to address the fundamental issue of insufficient bank liquidity while enhancing the overall quality and efficiency of financial services provided to the real economy.

Secondly, it is posited that the digital transformation of banks can mitigate liquidity shortages by optimizing loan loss provisions, enhancing risk tolerance, and curbing financial disintermediation. Commercial banks should leverage digital technology to reconstruct their financial logic and transform service models in order to bolster their risk resilience. Simultaneously, banks must personalize customer service and attract new clients through digital

channels to strengthen the internal capacity of their digital transformation efforts in serving the real economy.

Finally, given the heterogeneous effects of banking digitalization on liquidity creation, regulatory authorities should implement differentiated measures. They should extend support to rural and commercial banks that are proximate to entities and possess limited resources. This support aims to accelerate digital reform through targeted subsidies for financial technology and tax credits for research and development expenses incurred by small and medium-sized banks. Additionally, authorities should relax documentation requirements to facilitate the seamless integration of credit operations with digital technology. The objective is to support the real economy by lowering entry barriers. In regions experiencing significant growth in digital finance, banks are likely to encounter intensified competition from other financial institutions during their digital transformation processes. Consequently, there exists a potential shift away from credit-based operations. It is essential for regulators to continuously enhance digital regulatory frameworks and risk management platforms while concurrently establishing a regulatory framework that aligns with the banking sector in China.

## Author contributions

**Conceptualization:** Wen Wen.

**Data curation:** Wen Wen, Ying Liang.

**Funding acquisition:** Wen Wen.

**Investigation:** Ying Liang.

**Methodology:** Wen Wen.

**Writing – original draft:** Wen Wen.

**Writing – review & editing:** Ying Liang.

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
