## [Decision Letter · Decision Letter 0]

30 Jul 2024

Dear Dr. wen,

**Comments from PLOS Editorial Office**

We look forward to receiving your revised manuscript.

Kind regards,

Islam Abdeljawad

Academic Editor

PLOS ONE

Journal Requirements:

"1.Shaanxi Provincial Department of Education Special Research Program project "Digital Finance and traditional finance Coupling Empower Shaanxi Provincial Rural Revitalization Strategy Development" (project No. : 23JK0247); 2.Xianyang Normal University Special Research Fund Project "Research on Digital Financial Resource Allocation Mechanism in the Upgrading of Industrial Structure in Shaanxi Province" (Project No. : XSYK22010)"

Reviewers' comments:

Reviewer's Responses to Questions

**Comments to the Author**

1. Is the manuscript technically sound, and do the data support the conclusions?

Reviewer #1: Yes

Reviewer #2: Yes

Reviewer #3: Yes

2. Has the statistical analysis been performed appropriately and rigorously?

Reviewer #1: Yes

Reviewer #2: Yes

Reviewer #3: Yes

3. Have the authors made all data underlying the findings in their manuscript fully available?

Reviewer #1: Yes

Reviewer #2: Yes

Reviewer #3: Yes

4. Is the manuscript presented in an intelligible fashion and written in standard English?

Reviewer #1: Yes

Reviewer #2: No

Reviewer #3: Yes

Reviewer #1: Please follow the attachment for making your paper ready for publication. this will make it a good contribution in the selected area, overall its good work. Case studies can provide invaluable insights into real-world examples of digital entrepreneurship. Consider including case studies that illustrate successful (and perhaps unsuccessful) digital ventures to enhance the paper’s practical relevance. Digital entrepreneurship is not limited by geographical boundaries. Including a discussion on how digital entrepreneurship differs across various regions and cultures could add depth to the paper.

Reviewer #2: The subject is very interesting, and the database is recent, but the article cannot be published in its current form. Devoting 12 pages to the introduction is unjustifiable. I noticed much redundancy. In addition, the bibliographical references need to be updated. The discussion section should be linked to the theoretical part and highlight the implications of this study.

The last heading should read ‘conclusion’ only.

It's essential to thoroughly proofread your text to ensure it is free of grammatical or spelling errors, which can significantly enhance the readability and credibility of your work.

Reviewer #3: Dear authors,

It's my luck to review this paper, which focus on the digital transformation of commercial banks in China. It think this paper contribute greatly to the digital transformation process of commercial banks in emerging markets. The following constructive comments could be helpful for authors to develop the manuscript.

1. The introduction should mention the motivations to conduct this study in China, as prior studies such as Jiang Hai et al., 2023), Guo Ye et al., 2022), (Li Yifei et al., 2022) address the digital transformation index in China. What is the unique institutional characteristics about Liquidity creation and digital transformation in China?

2. The contribution needs citations and explain how this study extends prior studies in data, method and idea.

3. The literature should incorporate the theories that explain the nexus between digital transformation and liquidity creation. For example: Vuong, G. T. H., Phan, P. T. T., Nguyen, C. X., Nguyen, D. M., & Duong, K. D. (2023). Liquidity creation and bank risk-taking: Evidence from a transition market. Heliyon, 9(9).

4. The variable definitions should include variable definition tables to summary how authors calculate the variables.

5. In the baseline models, the authors employ the FEM method, how could the results are free from autocorrelation and heteroskedasticity issues?

6. Honestly, I loose my focus as the study incorporate so many objectives. It would be better for readers if the authors narrow down the focus and consolidate the content of the study.

7. The findings needs in-depth and economic discussion, rather than describe the tables.

8. There are many interesting results and the implications should closely linked with the main findings.

I hope the comments would are helpful and good luck to your publications.

**Do you want your identity to be public for this peer review?** For information about this choice, including consent withdrawal, please see our Privacy Policy

Reviewer #1: **Yes: ** Dr. Shujahat Ali

Reviewer #2: No

Reviewer #3: **Yes: ** Khoa Dang Duong

---

## [Author Response · Author response to Decision Letter 0]

12 Sep 2024

Reviewer 2#

Dear Editors and Reviewers:

Thank you very much for taking your time to review and comments concerning our manuscript entitled “Digital transformation and liquidity creation in commercial banks: evidence from the Chinese banking industry” . Those comments are all valuable and very helpful for revising and improving our paper. We have studied comments carefully and have made correction and point-by-point response which we hope meet with approval. Our responses are detailed below and highlighted in red. The main changes in the paper and the responds to the reviewer #2 are as follows:

Q1: The subject is very interesting, and the database is recent, but the article cannot be published in its current form. Devoting 12 pages to the introduction is unjustifiable. I noticed much redundancy. In addition, the bibliographical references need to be updated. The discussion section should be linked to the theoretical part and highlight the implications of this study.

Response: Indeed, the discussion on the limitations of the study was ignored. We have deleted some unnecessary phrases in the paper and modified some expressions in the paper.

First, in the introduction part, the language is simplified, and the entry point is found directly through the existing problems.

Second, we deleted some less relevant documents in Chapter 2 and added some new ones.

“In recent years, the continuous integration of digital technology represented by big data, cloud computing, and artificial intelligence with the financial industry has reshaped and promoted the innovation of financial products and service models, resulting in major changes in the business behavior and structure of banks (Liu, 2016; Guo and Zhu, 2021). These changes have had a profound impact on the level of liquidity creation within banks and their impact on the real economy (Song et al., 2023).” （LINE169-174）

“Meanwhile, liquidity creation is positively correlated with economic output and growth (Davydov et al., 2021), and the expansion of liquidity creation further promotes the growth of the real economy.” （LINE265-267）

Third, to make the research question clearer, we re-adjusted the mechanism of action diagram.

Q2: The last heading should read “conclusion” only.

Response: Thanks for this great point.

First, we have corrected the title of the last chapter.

Secondly, we have revised the research recommendations in the last chapter.

“First, the shortage of bank liquidity has hindered the transmission of monetary policy to a certain extent, and the digital transformation of banks can help improve the level of bank liquidity. Therefore, regulatory authorities should encourage commercial banks to promote the process of digital transformation and promote the deep integration of fintech, commercial banks and the real economy through policy support. (LINE 851-855)

Secondly, it is considered that the digital transformation of banks can alleviate the liquidity shortage by optimizing loan loss provisions, improving risk tolerance and restraining financial disintermediation. Commercial banks should make use of digital technology to reconstruct financial logic and transform service mode to improve their risk resistance ability. At the same time, banks need to personalize customer service and develop new customers through digital means, to improve the internal power of banks' digital transformation to serve the real economy. (LINE 875-880)

Finally, due to the heterogeneity of the effects of banking digitalization on liquidity creation, regulatory authorities should respond with differentiated measures.” (LINE 881-882)

This letter is our point-by-point response to the comments raised by the reviews. The comments are produced and our responses are given directly afterward in a different color from the authors in the revised copy.

We would also like to thank you for allowing us to resubmit a revised copy of the manuscript.

We hope that the revised manuscript may address the concerns raised by the reviewers and will have the opportunity to be published in PLOS ONE.

Response to Reviewer 3 Comments

Dear Editors and Reviewers:

Thank you very much for taking your time to review and comments concerning our manuscript entitled “Digital transformation and liquidity creation in commercial banks: evidence from the Chinese banking industry” . Those comments are all valuable and very helpful for revising and improving our paper. We have studied comments carefully and have made correction and point-by-point response which we hope meet with approval. Our responses are detailed below and highlighted in red. The main changes in the paper and the responds to the reviewer #3 are as follows:

Q1: The introduction should mention the motivations to conduct this study in China, as prior studies such as (Jiang Hai et al., 2023), (Guo Ye et al., 2022), (Li Yifei et al., 2022) address the digital transformation index in China. What are the unique institutional characteristics about Liquidity creation and digital transformation in China?

Response: We thank reviewer for reminding us this important point. We add the liquidity problem faced by China's financial market, which is very different from that of other countries, to introduce the necessity of the integration of digital technology and traditional finance and the main issues studied in this paper. The details are as follows:

“However, for a long time, China's real economy has been faced with financing difficulties and high financing costs, while the problem of idling arbitrage in the banking system still exists. In response to this problem, in recent years, the regulatory authorities have issued a few policies, such as making banks that meet the assessment requirements of inclusive finance enjoy a certain preferential deposit reserve ratio, reducing excess deposit reserves to release signals guiding credit expansion, etc. These measures have alleviated the shortage of funds in the real economy in the short term, but this problem has always been difficult to be substantively resolved.” (LINE 64-71)

Q2: The contribution needs citations and explain how this study extends prior studies in data, method and idea.

Response: Thank you very much for this comment. We have revised the innovation part of this paper as follows.

“Firstly, existing literature has begun to pay attention to the impact of digital transformation of banks on banking operations and the real economy, but the investigation on this issue focuses on the asset side of banks. The index of bank liquidity creation is a comprehensive measure of the smooth conduction of liquidity through the banking system by the asset side and the liability side of the bank. Therefore, from the perspective of liquidity creation, it is more comprehensive to examine the issue that the digital transformation of banks can help improve the efficiency of banks to serve the real economy. Secondly, the existing literature on the connection between digital transformation and mobility creation needs to be supplemented, both theoretically and empirically. This paper focuses more comprehensively on the potential for banks' digital transformation to reduce or increase liquidity creation through multiple transmission channels. At the theoretical level, the paper makes a detailed analysis of how the digital transformation of banks affects liquidity creation, and at the same time, it makes a more detailed test of the internal relationship between the two through empirical research. Thirdly, considering that there are considerable differences in the development of different types of banks and different regional banks in China's banking system, based on the development status of Chinese banks, this paper reveals the impact of digital transformation on liquidity when banks face different external environments and different internal conditions. It provides reliable empirical evidence for banks to formulate digital transformation strategies in accordance with local conditions in China.” (LINE 114-132)

Q3: The literature should incorporate the theories that explain the nexus between digital transformation and liquidity creation. For example: Vuong, G. T. H., Phan, P. T. T., Nguyen, C. X., Nguyen, D. M., & Duong, K. D. (2023). Liquidity creation and bank risk-taking: Evidence from a transition market. Heliyon, 9(9).

Response: We sincerely appreciate the valuable comment. In the second chapter, literature review and research hypothesis are reviewed, and the relationship between digital transformation and bank liquidity creation is directly emphasized. Specifically as follows：

“In recent years, the continuous integration of digital technology represented by big data, cloud computing and artificial intelligence with the financial industry has reshaped and promoted the innovation of financial products and service models, resulting in major changes in the business behavior and structure of banks (Liu, 2016; Guo and Zhu, 2021). These changes have had a profound impact on the level of liquidity creation within banks and their impact on the real economy (Song et al., 2023).” (LINE 164-169)

Q4: The variable definitions should include variable definition tables to summary how authors calculate the variables.

Response: Thanks for this comment. We added the measurement of variables in the descriptive statistics table to make the definition of variables more intuitive. At the same time, the profitability statistics are supplemented.

Variable symbol Method of measurement Mean value Standard deviation Minimum value median Maximum value

liquidity creation Lc Constructed according to the three-step cat-fat classification method proposed by Berger and Bouwman (2009) 54.345 17.421 24.078 51.436 192.160

Digitization of Banking Services Tdi Peking University Digital Transformation Index of Chinese commercial banks 52.137 38.468 0 52 200

Provision for loan losses Llp Loan loss reserve/total loans 0.033 0.937 0 0.051 0.152

Risk tolerance Risk Risk-weighted assets/total assets 0.587 5.342 0.223 0.649 0.915

Financial disintermediation Findis (regional corporate bonds + non-financial corporate stocks)/social financing scale 0.156 3,552 0.375 0.147 0.08

Industrial structure Ind Gross secondary industry product/Gross regional product 46.743 9.451 32.762 48.702 81.328

Business cycle M2 M2 growth rate 11.460 2.346 6.990 11.010 18.950

Level of regional economic development Gdpg Regional GDP growth rate 10.768 0.654 9.299 10.739 12.123

Non-performing loan ratio Npl Non-performing loans/total loans 1.627 0.764 0.049 1.214 4.376

Capital adequacy ratio Tcr Total capital/risk-weighted assets 13.449 3.135 0.018 12.771 57.610

Asset size Size Ln (Total assets) 25.674 1.529 22.391 25.395 31.138

Percentage of non-interest income Nir Non-interest income/operating income 34.216 10.159 0 33.315 206.077

Profitability Roe Net profit/net assets 15.172 6.383 0.813 13.624 32.469

Q5: In the baseline models, the authors employ the FEM method, how could the results are free from autocorrelation and heteroskedasticity issues?

Response: Thanks for this comment. To control heteroscedasticity and autocorrelation problems, we Cluster all the standard errors of regression coefficients at the bank level. It is explained in P10.

“To control for potential heteroscedasticity and serial correlation, all regression coefficients are clustered and have robust standard errors.” (LINE 460-461)

Q6: Honestly, I loose my focus as the study incorporate so many objectives. It would be better for readers if the authors narrow down the focus and consolidate the content of the study.

Response: Thanks for this great point. We have deleted some unnecessary phrases and other descriptions in the paper. For example, first, we simplify the language in the introduction and find the entry point directly through existing problems; (P2-5)

Second, in the empirical results part, I asked to test the analysis part of each table and analyze the empirical results combined with the economic reality; (P21-36)

Third, to make the research question clearer, we re-adjusted the mechanism of action diagram.

Q7: The findings needs in-depth and economic discussion, rather than describe the tables.

Response: Indeed, the discussion on the limitations of the study was ignored. In the table test analysis part of the empirical results, we analyze the empirical results combined with the economic reality to help understand the research problem in this paper.

“Column (1) only considers the digital transformation of banks as a regression variable, and column (2) adds control variables. The results show that there is a significant positive correlation between digital transformation and liquidity creation, and indicates that the increase of bank liquidity creation is related to the process of digital transformation of banks. Our results are consistent with existing research demonstrating that the integration of digital technology and finance can help banks expand long-tail customers (Zhang and Naceur, 2019), increase loan volume (Sheng, 2021) and increase bank credit supply, thereby increasing bank liquidity creation. So, hypothesis 1 is tested.” (LINE 486-493) (LINE 64-71)

“Due to the application of digital technology, commercial banks have improved the ability to monitor loan credit risks, improved the forward-looking provision for credit losses, and avoided excessive provision for loan losses to cushion risks.” (LINE 546-548)

“This is due to the application of advanced technology in bank business activities, which effectively improves the information transparency of lending enterprises and reduces banks' concerns about unexpected risks. Therefore, under the premise of controllable risks, banks tend to obtain more interest spread returns by taking certain risks voluntarily, which is ultimately reflected in the increase of risk tolerance.” (LINE 565-569)

“Because banks use digital technology to lower the threshold of financial services, expand loan services for small and micro enterprises, agriculture and vulnerable groups, ease the financing constraints of these groups, and promote the development of the real economy.” (LINE 586-589)

“The possible reason is that in regions with perfect digital infrastructure, banks can take advantage of superior conditions to promote their own digital transformation, and advance the use of advanced technology to innovate financial products to enhance their competitiveness, thus promoting the creation of bank liquidity.” (LINE 718-721)

“This suggests that the development of digital finance will weaken the positive relationship between the digital transformation of banks and the creation of liquidity, which may be because the development of digital finance, while improving the efficiency of financial services and stimulating the digital transformation of commercial banks, will give birth to more financial service platforms to seize the credit market, form a new competitive pattern, and reduce the credit market share of commercial banks.” (LINE 739-744)

“This may be because banks with high resource endowments typically have higher levels of capitalization, can facilitate the use of fintech for business transformation and product innovation, and have strong borrowing capacity, thus creating more liquidity (Fang et al., 2023).” (LINE 769-772)

“This may be because the credit structure of large state-owned banks and joint-stock banks shows the characteristics of concentrated customers, high proportion of large loans, and systematic business development. With the support of digital technology, large banks have more competitive advantages in marketing, risk assessment and other aspects. Reducing the operating cost of bank credit business through economies of scale is helpful to improve the willingness of banks to invest liquidity in the medium and long term, thus improving the level of liquidity creation (Xiang and Gao, 2023).” (LINE 788-794)

8. There are many interesting results and the implications should closely linked with the main findings.

Response: Thanks for this great point. In the conclusion part of the sixth chapter, we reorganize and put forward corresponding policy suggestions according to the research conclusions of this paper

“First, the shortage of bank liquidity has h

---

## [Decision Letter · Decision Letter 1]

9 Oct 2024

Digital transformation and liquidity creation in commercial banks: evidence from the Chinese banking industry

PLOS ONE

Dear Dr. wen,

Thank you for submitting your manuscript to PLOS ONE. After careful consideration, we feel that it has merit but does not fully meet PLOS ONE’s publication criteria as it currently stands. Therefore, we invite you to submit a revised version of the manuscript that addresses the points raised during the review process.

We look forward to receiving your revised manuscript.

Kind regards,

Islam Abdeljawad

Academic Editor

PLOS ONE

Reviewers' comments:

Reviewer's Responses to Questions

**Comments to the Author**

Reviewer #2: All comments have been addressed

Reviewer #3: All comments have been addressed

2. Is the manuscript technically sound, and do the data support the conclusions?

Reviewer #2: Yes

Reviewer #3: Yes

3. Has the statistical analysis been performed appropriately and rigorously?

Reviewer #2: Yes

Reviewer #3: Yes

4. Have the authors made all data underlying the findings in their manuscript fully available?

Reviewer #2: Yes

Reviewer #3: Yes

5. Is the manuscript presented in an intelligible fashion and written in standard English?

Reviewer #2: Yes

Reviewer #3: No

Reviewer #2: The corrections are well incorporated, so I have no further comments. I recommend acceptance of the article

Reviewer #3: Dear authors,

Many thanks for spending time revising the manuscript. I totally agree with your responses. I think the manuscript could be benefit from professional proofreading service.

Good luck to your publication,

Best regards,

**Do you want your identity to be public for this peer review?** For information about this choice, including consent withdrawal, please see our Privacy Policy

Reviewer #2: No

Reviewer #3: No

---

## [Author Response · Author response to Decision Letter 1]

26 Oct 2024

Thank you very much for this comment. First, we thank reviewer for reminding us this important point. We made some modifications to the expressions in the paper. We also sought the help of a professional language editor to conduct a further review and revision of the paper's language expression. Second, we re-examined and corrected the existing measurement models, and re-evaluated the mediation methods mentioned in the paper. At the same time, we considered and analyzed all possible scenarios that could arise in the mediation model. Finally, We have revised the introduction section of the paper to highlight the main theme and significance of the study more prominently. We have also modified the conclusion section to provide a more in-depth description of the study's focus and significance.

---

## [Decision Letter · Decision Letter 2]

25 Nov 2024

Digital transformation and liquidity creation in commercial banks: evidence from the Chinese banking industry

PLOS ONE

Dear Dr. wen,

Thank you for submitting your manuscript to PLOS ONE. After careful consideration, we feel that it has merit but does not fully meet PLOS ONE’s publication criteria as it currently stands. Therefore, we invite you to submit a revised version of the manuscript that addresses the points raised during the review process.

Reviewer 4 has a concern arises regarding the decision to exclude only the year 2020 to mitigate the impact of the COVID-19 pandemic on the robustness tests. Given that the effects of the pandemic persisted until 2022, the rationale for this exclusion warrants further clarification.

We look forward to receiving your revised manuscript.

Kind regards,

Islam Abdeljawad

Academic Editor

PLOS ONE

Journal Requirements:

Reviewers' comments:

Reviewer's Responses to Questions

**Comments to the Author**

Reviewer #3: All comments have been addressed

Reviewer #4: All comments have been addressed

2. Is the manuscript technically sound, and do the data support the conclusions?

Reviewer #3: Yes

Reviewer #4: Yes

3. Has the statistical analysis been performed appropriately and rigorously?

Reviewer #3: Yes

Reviewer #4: Yes

4. Have the authors made all data underlying the findings in their manuscript fully available?

Reviewer #3: Yes

Reviewer #4: Yes

5. Is the manuscript presented in an intelligible fashion and written in standard English?

Reviewer #3: Yes

Reviewer #4: Yes

Reviewer #3: Dear authors,

I would like to thanks you for spending time addressing all my comments. Good luck to your publication.

Reviewer #4: The topic is of significant interest, and the research problems are clearly articulated. Identifying research gaps is well-justified, and the literature review adequately supports the hypothesis formulation. However, a point of concern arises regarding the decision to exclude only the year 2020 to mitigate the impact of the COVID-19 pandemic on the robustness tests. Given that the effects of the pandemic persisted until 2022, the rationale for this exclusion warrants further clarification.

**Do you want your identity to be public for this peer review?** For information about this choice, including consent withdrawal, please see our Privacy Policy

Reviewer #3: No

Reviewer #4: No

---

## [Author Response · Author response to Decision Letter 2]

30 Nov 2024

We thank reviewer for reminding us this important point. Through empirical analysis and discussion, we believe that it is more appropriate to exclude the data of 2020 in this paper. For this, we offer the following explanation:

Firstly, 2020 was a crucial year for the outbreak of the COVID-19 pandemic, and it was also the year when society was most severely impacted. Due to lockdowns and isolation measures, China's supply chains were disrupted, consumer demand plummeted, and bank liquidity, asset quality, and credit demand experienced significant fluctuations. For example, China's GDP increased from 98.65 trillion yuan in 2019 to 101.36 trillion yuan in 2020, with economic growth slowing from 6% to 2.2%. Therefore, 2020 should be prioritized for exclusion.

Secondly, in 2021, the global economy continued to be affected by the fluctuations of the pandemic, but the economy and financial markets began to recover. For example, China's GDP in 2020 was 101.36 trillion yuan, and economic growth was severely stifled, with a mere 2.2% growth rate. In 2021, China's economic recovery trend was obvious, with GDP reaching 114.92 trillion yuan and an 8.4% growth rate. In 2022, the global economy also began to recover, and China gradually loosened economic and trade restrictions. At the same time, digital economy grew rapidly. The liquidity and credit demand of Chinese commercial banks began to approach normal levels. In 2022, the total amount of yuan loans increased by 21.31 trillion yuan, an increase of 136 billion yuan compared to the previous year. The impact of digital transformation on the economy may have become apparent in 2021-2022. Therefore, we suggest retaining the data of 2021 and excluding the data of 2020.

---

## [Decision Letter · Decision Letter 3]

22 Jan 2025

Digital transformation and liquidity creation in commercial banks: evidence from the Chinese banking industry

PONE-D-24-15838R3

Dear Dr. wen,

We’re pleased to inform you that your manuscript has been judged scientifically suitable for publication and will be formally accepted for publication once it meets all outstanding technical requirements.

Kind regards,

Islam Abdeljawad

Academic Editor

PLOS ONE

Additional Editor Comments (optional):

Reviewers' comments:

Reviewer's Responses to Questions

**Comments to the Author**

Reviewer #5: All comments have been addressed

2. Is the manuscript technically sound, and do the data support the conclusions?

Reviewer #5: Yes

3. Has the statistical analysis been performed appropriately and rigorously?

Reviewer #5: Yes

4. Have the authors made all data underlying the findings in their manuscript fully available?

Reviewer #5: Yes

5. Is the manuscript presented in an intelligible fashion and written in standard English?

Reviewer #5: Yes

Reviewer #5: (No Response)

**Do you want your identity to be public for this peer review?** For information about this choice, including consent withdrawal, please see our Privacy Policy

Reviewer #5: No

---

## [Editor Report · Acceptance letter]

PONE-D-24-15838R3

PLOS ONE

Dear Dr. Wen,

I'm pleased to inform you that your manuscript has been deemed suitable for publication in PLOS ONE. Congratulations! Your manuscript is now being handed over to our production team.

Kind regards,

on behalf of

Dr. Islam Abdeljawad

Academic Editor

PLOS ONE